# OpenReview forum: "Scalable Extraction of Training Data from Aligned, Production Language Models"
_ICLR.cc/2025/Conference — ICLR 2025 Poster_

### Official Review · Reviewer_JxaE · 2024-11-03

**Soundness:** 3
**Presentation:** 2
**Contribution:** 2
**Rating:** 6
**Confidence:** 4

**Summary:**

The paper conducts a large amount of empirical research and finds that aligned chat models hardly leak training data. However, when the authors implemented the divergence attack and fine-tuning attack, the models leaked some training data, demonstrating significant security vulnerabilities in current large language models. The paper conducted a large number of experiments to validate the various negative effects on the model after being attacked.

**Strengths:**

- The paper conducted a large number of experiments to reveal the extraction attacks faced by large language models, and the models used in the experiments are very representative.
- The research problem addressed in the paper is very interesting; extraction attacks are an important topic for large language models.
- The structure of the paper is very well-organized, with rich details such as explanations and definitions for memorization, making it easy to read.

**Weaknesses:**

- I think that the paper lacks innovation or technical contribution. Although the two attack methods proposed in the paper reveal security issues with large language models, I think such contributions may not be sufficient for a top conference like ICLR.
- The divergence attack proposed in the paper is intriguing, but why does this attack work? Under what circumstances does it work? It seems that this attack may not enable targeted attacks (i.e., leaking specific information from the model). There appears to be a significant random component, which means that the efficiency of this type of attack may be low for the attacker.
- It seems that the authors did not discuss the relationship with related works. Some adversarial attacks also seem to achieve similar effects. What are the main differences between the authors' work and related works?

**Questions:**

Please refer to the ``Weaknesses`` part.

---

> ### Author Response · Authors · 2024-11-18
>
> Thank you for your review and valuable feedback. We address the specific weakness and questions next:
>
> > I think that the paper lacks innovation or technical contribution.
>
> We disagree and believe the innovation and contributions of this work are as follow:
>
> * This is the **first work demonstrating that alignment can be circumvented to exfiltrate training data** at very high rates. This had not been discovered for a year after the release of the first ChatGPT, and OpenAI believed it was not possible. This highlights our technical contribution is not trivial and has had an impact in system design and increased the privacy of state-of-the-art models, as discussed in our responsible disclosure section.
> * Showcasing **limitations of alignment** is useful for the larger community understanding how to improve the reliability of state-of-the-art models.
>
> > The divergence attack proposed in the paper is intriguing, but why does this attack work?
>
> The nature of the divergence attack is indeed intriguing as we motivate in the paper. Since we do not have specific knowledge about GPT-3.5, it is hard to determine the reasons for this to happen. These motivates our fine-tuning attack that presents a more principled approach to bypass alignment, achieves higher success rates, and enables the attack to extract specific documents.
>
> > It seems that the authors did not discuss the relationship with related works. Some adversarial attacks also seem to achieve similar effects. What are the main differences between the authors' work and related works?
>
> This work is the first to introduce empirical attacks that can bypass alignment to extract memorized data from aligned, production models. Do you have specific pointers to similar attacks and related works that are not already mentioned in Section 2 that could improve the contextualization?

---

> > ### Comment · Reviewer_JxaE · 2024-11-22
> >
> > Thank you for your response. Your first point makes me reconsider the contributions of the paper.
> >
> > However, the second point does not convince me, and I still have doubts about the method proposed in the paper. While it achieves certain results, I do not understand the motivation and principles behind the method. How is the trigger prompt for the DIVERGENCE attack designed? Is it determined through experimentation? Are there other forms of this trigger prompt? If there is only this one form, wouldn't it be easy to defend against? Does this mean that the attack lacks generalizability?
> >
> > If you have any other points that you haven't articulated yet, feel free to discuss them with me further.

---

> > > ### Author Response · Authors · 2024-11-22
> > > **response**
> > >
> > > We thank the reviewer for reconsidering our contributions in light of our rebuttal.
> > >
> > > Regarding the divergence attack, let us try to give some additional context that may be helpful.
> > >
> > > When we started this work, fine-tuning was not available for most proprietary models.
> > > So the only way to try and elicit memorization was through prompting.
> > > But it is really unclear how one would go about finding or optimizing a prompt that leads to training data regurgitation (compared to, say, finding prompts for jailbreaks).
> > >
> > > We actually originally found the divergence attack while studying a different question, of how models behave when they exhaust their context window (an easy way to do this was to ask the model to repeat a word forever).
> > > So in a sense the reviewer is right that there is little guiding principle behind this attack. But the attack in itself is not what we find most interesting since it is easily patched and doesn't seem to generalize well to other models (as some reviewers have noted).
> > >
> > > Rather, we view this attack as a very clear demonstration of the difficulty of red-teaming models for unwanted behaviors. Because of this, it took over a year after ChatGPT was released before this attack vector was found. Now that it is patched, we are confident that other, similar attack vectors exist. But it remains entirely unclear how to find them efficiently (so, it is likely that if other prompt attacks are discovered in the future, it will also be "by chance"). This is a terrible outlook from a security perspective, and our work aims to highlight this.
> > >
> > > But now, we also have the ability to finetune some models (e.g., GPT-3.5 and GPT-4). And here, we show that a very principled, successful and generalizable attack does exist. So another lesson from this work is that red-teamers should focus more on finetuning to efficiently discover model vulnerabilities (and therefore, it is valuable if model providers give access to finetuning to security researchers).
> > >
> > > We hope this additional context helps understand the motivations behind our work and our different attacks.
> > > We will aim to include this context in a camera-ready version of our paper upon acceptance (some extra details we can provide might violate the double-blind policy).

---

> > > > ### Comment · Reviewer_JxaE · 2024-11-22
> > > >
> > > > Thank you for your analysis. I reconsidered the work of the paper, and although I believe the technical contributions are not particularly outstanding, I partially acknowledge the other contributions (some key experimental conclusions) of the paper. Therefore, I have increased my rating to 6.

---

### Official Review · Reviewer_9CXv · 2024-11-04

**Soundness:** 4
**Presentation:** 2
**Contribution:** 3
**Rating:** 6
**Confidence:** 3

**Summary:**

This paper considers two approaches to extract a large language model's training data. The first is using repeated words, and the second is by fine-tuning the model to break the safety training. The approach is tested on proprietary models, and shown to regenerate sentences from the open source datasets with verbatim tokens over a threshold. While the first approach does not always work, fine-tuning could easily circumvent the defense mechanism put in the model.

**Strengths:**

- S1: The approaches are simple and effective without too many assumptions.
- S2: The attacks are shown to work on the state-of-the-art commercial models.
- S3: The presentation is good overall and the paper is very easy to read.

**Weaknesses:**

- W1: The related work section is missing. Although there is the background section, the paper does not properly cover the related work and its relation to the existing work, as well as potential defenses in the literature. Especially, these attacks are known and discussed in different forums. There is a potential that the authors of the paper might be those who suggested and discussed these approaches early on, but some mention of the context is useful understanding the literature and the significance of this approach.
- W2: The paper defers a lot of information to the appendix. Although this abundance of information comes from the thorough analysis and investigation, the paper needs to prioritize more essential information and drop potentially duplicate or obvious information.

**Questions:**

- Q1: What are the examples of the (near) duplicate generations and their significance?

The paper is overall well written and the extensive analysis is helpful, the paper can be improved with better use of prioritization in space, and a proper related work section. Especially, discussing how novel the proposed approach would be helpful understanding the impact of this paper. This might need a significant reorganization of the paper, but all the ingredients should be already there. If that can be done, I'm willing to upgrade my recommendation.

---

> ### Author Response · Authors · 2024-11-18
>
> Thank you for your review and valuable feedback. We address the specific weakness and questions next:
>
> > The related work section is missing.
>
> It is not. Section 2 covers related work and necessary background.
>
> > These attacks are known and discussed in different forums.
>
> Yes, we disclosed our work a while back following our responsible disclosure with OpenAI, and we are aware this work has been discussed. We do not include mentions to preserve anonymity.
>
> > The paper defers a lot of information to the appendix.
>
> We believe the main paper is self-contained including most relevant information for our main contribution “alignment is not enough to prevent regurgitation”. We obtained many observations along the way that we think is worth reporting in the Appendix. Do you have specific comments regarding what information is redundant or how presentation of this information could be improved?
>
> > What are the examples of the (near) duplicate generations and their significance?
>
> We do not have such examples as they cannot be captured by our deterministic method. We mention that this is important since our results will not capture (near) duplicates, which may also be problematic in some scenarios. You could think of (near) duplicates as the same sentence replacing one adjective by a synonym, or using commas instead of parentheses.

---

> > ### Comment · Reviewer_9CXv · 2024-11-20
> >
> > The main problem with the comingled background section instead of the related work is that there are many distractions and the training data extraction appears only as part of "Prompts" paragraph, without going into the similarity and difference to the proposed approach. More details here will be helpful understanding the position of this paper in the literature.
> >
> > Thank you for clarifying about the prior disclosure. I did like the idea, and I believe it makes sense to formally disseminate the findings. So, I'm upgrading my recommendation. This decision comes without my remembering the actual authors or the affiliation to introduce a bias and after recognizing this work was not a copycat of a known attack.
> >
> > By the way, there's a minor typo in Line 71.

---

### Official Review · Reviewer_tvra · 2024-11-04

**Soundness:** 4
**Presentation:** 4
**Contribution:** 3
**Rating:** 8
**Confidence:** 4

**Summary:**

This paper pointed out the key reasons of the ineffectiveness of the model alignment and developed two novel techniques to circumvent chatbot alignment guardrails: a divergence attack and a finetuning attack.  The author demonstrated that this is the first large-scale, training-data extraction attacks on proprietary language models using only publicly-available tools and relatively little resources. This work highlights the limitations of existing safeguards to prevent training data leakage in
production language models. And the experiment results show the model alignment is not enough to prevent memorization.

**Strengths:**

1. The contributions are valid and significant. This work highlighted the limitations of existing safeguards to prevent training data leakage in
production language models. The author proposed two novel extraction attacks illustrating the limitation of model alignment of training-data extraction. The attacks only require access to tools that are publicly accessible to everyone. In addition, the author proposed a scalable approach to validate memorization.

2. The paper does a comprehensive research showing additional work in long Appendix with sufficient experiments.

3. The paper has good structure by clarifying key definitions and prompting the motivation. In experiments, the author clearly described the scalable approach for validating memorization and what are the production language models, including both aligned, conversational models and instruction-tuned, non-conversation models.

**Weaknesses:**

1. (not a weakness but a suggestion). During the reading, I found some figures and conclusions in the Appendix is helpful and may worthwhile to be added or replaced to the main body. For example, Figures in Appendix A.9

2. In section 7 QUALITATIVE ANALYSIS OF EXTRACTED TEXT, it seems the result analysis focuses on the length of the extracted string and memorized text. It may better if the author could add more explanation in terms of the leakage of random training data from divergence attack vs the leakage of specification training data from fine-tuning attack.

**Questions:**

1. [line 047] It said they apply divergence attack to ChatGPT and Gemini but apply finetuning attack to ChatGPT only. Is there a particular reason why they doesn’t apply finetuning attack to Gemini?

---

> ### Author Response · Authors · 2024-11-18
>
> Thank you for your review and valuable feedback. We address the specific weakness and questions next:
>
> > In section 7 QUALITATIVE ANALYSIS OF EXTRACTED TEXT, it seems the result analysis focuses on the length of the extracted string and memorized text. It may better if the author could add more explanation in terms of the leakage of random training data from divergence attack vs the leakage of specification training data from fine-tuning attack.
>
> Thank you for bringing this up. Analyzing these pieces of text automatically is quite hard, let us know if you have specific metrics or analysis that could help. Nevertheless, our targeted fine-tuning is a qualitative different way of extracting data as we only count an attack successful if it extracts exactly the next tokens in the document.
>
> > During the reading, I found some figures and conclusions in the Appendix is helpful and may worthwhile to be added or replaced to the main body. For example, Figures in Appendix A.9
>
> Thank you for your suggestion. Unfortunately, space limitations do not allow us to include further details.
>
> > [line 047] It said they apply divergence attack to ChatGPT and Gemini but apply finetuning attack to ChatGPT only. Is there a particular reason why they doesn’t apply finetuning attack to Gemini?
>
> Gemini does not provide a public fine-tuning API as OpenAI does. It is thus not possible to finetune Gemini.

---

> > ### Comment · Reviewer_tvra · 2024-11-22
> >
> > Thanks for the response. I will keep my rating to endorse this paper's quality.

---

### Official Review · Reviewer_4xM4 · 2024-11-04

**Soundness:** 3
**Presentation:** 2
**Contribution:** 3
**Rating:** 6
**Confidence:** 4

**Summary:**

This paper compares pretrained base models and aligned production models using a simple completion attack to extract training data. The findings indicate that the alignment process appears to prevent models from directly outputting training data when faced with this straightforward attack.
To bypass the defense mechanisms introduced by alignment, the paper proposes two novel techniques for extracting training data from aligned production LLMs: the divergence attack and the fine-tuning attack. In the divergence attack, the model is prompted to perform a repetitive task, such as repeating a specific word. This can lead the model to deviate from the original task and potentially output training data. The fine-tuning attack involves fine-tuning the model with a completion task similar to the initial completion attack, using a set of 2,000 data points.
To quantitatively assess the effectiveness of these techniques, a 10TB text dataset was constructed as the ground truth for training data comparison. The results demonstrate that the divergence and fine-tuning attacks were able to extract training data from ChatGPT at rates of 3% and 23%, respectively.
In addition to extracting training data, these attacks also induced the model to produce harmful content.

**Strengths:**

- This paper underscores an important problem that current alignment techniques do not fully mitigate risks of extracting training data from LLMs.
- This paper demonstrates the successful extraction of training data from production models in significant quantities and at a feasible cost.
- This paper introduces a large dataset and a searching algorithm to act as a proxy for unknown training datasets and help matching the data.

**Weaknesses:**

- The divergence attack causes the model to output training data as part of its response. An additional step is needed to compare different parts of these responses with the training dataset to verify extraction. The success rate of these attacks remains limited as well. These show a gap between successfully extracting unknown training data and performing an attack similar to the membership inference attack.
- While testing baseline attacks on 9 open base models, the paper only tests baseline attacks on one aligned model, GPT-3.5. It requires testing on more aligned models to support the claim.
- The divergence attack proves effective only on ChatGPT and does not transfer to other models, such as Gemini.
- The finetuning attack has been evaluated solely on LLaMA-2-chat and ChatGPT, despite the existence of many new aligned open-source models that could be used to further assess the attack's effectiveness. The results from LLaMA-2-chat indicate limited effectiveness and the transferability limitations of the attack.

**Questions:**

- Conduct more experiments for baseline attacks on more aligned models to support the conclusion---"Baseline attacks fail against aligned models".
- Conduct more comprehensive experiments with more and newer models for the finetuning attack.
- Estimate the probability of extracting the training data part from the whole response assuming the training data is unknown.
- Minor problems:
  - line 071: the broken symbol before "10,000 examples"
  - Figure 2 is never mentioned in the main text.

---

> ### Author Response · Authors · 2024-11-18
>
> Thank you for your review and valuable feedback. We address the specific weakness and questions next:
>
> > Membership inference attack is required to verify if extracted data is in training.
>
> A membership inference attack would indeed be needed if the model had a completely private training dataset. In this work, however, we use the prior knowledge we have of models being trained on the entire Internet. This allows us to approximate the private training set with the collection of public datasets compiled in AuxDataset and **exactly** compute what amount of generated text that can be found online. This overlap will provide us with a lowerbound on the memorized and extracted data. Also, since the snippets we consider are quite long, we expect the rate of false positives to be virtually zero as there is very little probability of a model producing them by chance without memorizing it during training.
>
> > Paper only tests attacks on one aligned model, GPT-3.5. It requires testing on more aligned models to support the claim.
>
> Thank you for bringing this up. We have discussed this in our general response. Our research first uncovered the divergence attack on GPT-3.5. We disclosed this following responsible disclosure best practices to OpenAI, as detailed in the paper. OpenAI patched this vulnerability for this model and all future models. Similar protections seem to have been applied in models from different providers.
>
> We evaluate our fine-tuning attack on several aligned models (Llama, GPT-3.5 and GPT-4) with similar results and demonstrating transferability.
>
> > Results of fine-tuning attack on LLaMA-2-chat indicate limited effectiveness and the transferability limitations of the attack.
>
> We have also discussed this in our general response. Our attack is effective for Llama-2 but memorization rates are not directly comparable to OpenAI models. If you take a close look at Table 2, the extracted tokens for Llama 2 **before alignment** are below 4% compared to 16% for aligned GPT-3.5 **after attack**. This means that Llama 2 has memorized less text to begin with. Still our attacks are the best known to date and can get over 30% of the memorized text from aligned models.
>
> > Conduct more comprehensive experiments with more and newer models for the finetuning attack.
>
> Thank you for the suggestion. We have decided not to conduct further experiments as the current ones already demonstrate our main thesis “alignment is not enough to prevent adversarial extraction of pre-training data”.
>
> > Estimate the probability of extracting the training data part from the whole response assuming the training data is unknown.
>
> As we discuss above, AuxDataset provides us with a very good approximation of the training data. Studying the effectiveness of membership inference attacks on a completely private training set is orthogonal to our goals, and could be explored by future work.

---

> ### Comment · Reviewer_4xM4 · 2024-11-22
>
> Thanks for the response! It addressed most of my questions.
> But I am still wondering about the comparison between training data extraction and membership inference. For example, in the divergence attack, the approach extracts 1,000 tokens, but only several chunks of tokens (~50 tokens) are training data. Currently this work verifies the training data by comparing with a proxy AuxDataset, but lacking the discussion about what if the training data is unknown, the probability of accurately extracting the chunks is much lower. The success rate in this case is different from the success rate in the finetune attack, which extracts the training data starting from the first several tokens in order.

---

> > ### Author Response · Authors · 2024-11-22
> > **response**
> >
> > Thanks for the insight, this is a distinction between the two attacks we hadn't considered.
> >
> > Since the training data is publicly available here, our goal is primarily to quantify how much training data is leaked by the model. This sets an upper bound on the amount of data that an attacker could confidently extract from the model.
> >
> > As we see it, there are two options an attacker could use to get a confident attack:
> > 1) membership inference, as the reviewer suggests. We view these attacks as complimentary to our work: any improvement in membership inference will translate to a more precise data extraction attack. And so we leave this out-of-scope.
> > 2) for many types of sensitive data, there are out-of-bands ways for an attacker to verify whether the data is correct. For example, suppose an attacker extracts a name and social security number, and doesn't know if this is real training data or hallucinated. Rather than attempt a MI attack, the attacker could just check whether the SSN is accepted by some government service.
> >
> > We will clarify these points in the paper.

---

> > > ### Comment · Reviewer_4xM4 · 2024-11-25
> > >
> > > It makes more sense. As the authors will clarify these in the paper, I am raising my score. Thanks for the great work!

---

### Official Review · Reviewer_8SjV · 2024-11-04

**Soundness:** 3
**Presentation:** 3
**Contribution:** 3
**Rating:** 8
**Confidence:** 2

**Summary:**

This paper highlights that, despite alignment, large language models still have potential risks of leaking training data. The authors introduce two novel attack techniques, the divergence attack and the finetuning attack, to bypass alignment safeguards. The methods successfully extract thousands of data samples from models like OpenAI's ChatGPT and Google's Gemini.

**Strengths:**

Originality & Significance: This paper provides valuable insights into the limitations of current alignment methods in reducing the risk of training data extraction. The proposed extraction methods are both highly scalable and cost-effective.
Clarity: The paper is well-structured and easy to follow, with clear and detailed descriptions of the experiments.

**Weaknesses:**

The paper contains experimental details and some analysis of how model capacity influences memorization. The analysis is more empirical than theoretical and lacks a detailed theoretical examination of why model capacity correlates with memorization in this way.

**Questions:**

See weakness.

---

> ### Author Response · Authors · 2024-11-18
>
> Thank you for your review and valuable feedback. We address the specific weakness and questions next:
>
> > The paper contains experimental details and some analysis of how model capacity influences memorization. The analysis is more empirical than theoretical and lacks a detailed theoretical examination of why model capacity correlates with memorization in this way.
>
> Thank you for bringing this up. It is hard to establish theoretical grounding for these results. The main intuition we have is that larger models have more capacity to store information if trained on the same data compared to smaller models. We believe this exploration is beyond the scope of our work, and has been studied by other works such as [1,2].
>
> [1] Biderman, Stella, et al. "Emergent and predictable memorization in large language models." Advances in Neural Information Processing Systems 36 (2024).
>
> [2] Carlini, Nicholas, et al. "Quantifying memorization across neural language models." arXiv preprint arXiv:2202.07646(2022).

---

> > ### Comment · Reviewer_8SjV · 2024-12-03
> >
> > Thanks for your response. I will keep my original rating.

---

### Official Review · Reviewer_JkUJ · 2024-11-05

**Soundness:** 4
**Presentation:** 4
**Contribution:** 3
**Rating:** 6
**Confidence:** 4

**Summary:**

This paper provides a pioneering study on scaled evaluation of training data memorization issues in aligned Large Language Models (LLMs). The paper effectively defines memorization as the generation of at least 50 tokens that match training data. The authors created AUXDATASET, a 10-terabyte dataset merging four of the largest published language model training datasets, enabling systematic evaluation of the lower bound of training data memorization.

The study focuses on three aligned models (with 9 open-weight non-aligned models as baselines). GPT-3.5-Turbo/Gemini 1.5 Pro was primarily studied under prompt-based divergence attacks, while both GPT-3.5-Turbo and GPT-4, along with Llama-2-chat, were evaluated using fine-tuning-based divergence attacks to remove chatbot-like behaviors for better assessment.

The authors discovered that their divergence attacks (causing deviation from typical chatbot behavior) significantly increased the success rate of extracting memorized content from potential training data. Qualitatively, they identified memorization issues in OpenAI models, including OpenAI's proprietary data not released to the public, copyright-protected content from The New York Times, toxic content, and private information.

**Strengths:**

- The paper addresses a critical problem in LLM development with robust methodology. The authors establish a formal framework by providing a clear definition of memorization (i.e., >50 tokens), creating a comprehensive validation corpus, and presenting results as quantifiable lower bounds on memorization issues.

- The technical innovation in attack methods is compelling (**but might only be one correlated aspect of memorization as it seems the divergence attacks are solely effective to GPT models, see weakness**). The authors propose two effective approaches: a prompt-based method utilizing word repetition to elicit divergent behavior (**which seems to have been fixed by OpenAI**), and a more sophisticated fine-tuning-based divergence attack. Both methods successfully demonstrate how to bypass chatbot-like behaviors to expose memorization from OpenAI models.

- The empirical analysis is thorough and well-structured. The study reveals interesting correlations between memorization and model size and introduces meaningful metrics such as unique 50-grams for measurement. The large-scale evaluation of 10 terabytes of data provides robust evidence for their findings.

- The findings from OpenAI models are compellingly grounded in practical implications, demonstrating memorization of sensitive content including The New York Times' copyrighted material, toxic content, personally identifiable information (PII), and OpenAI's unreleased training data. This connection to real-world concerns enhances the paper's significance.

- The paper is well-structured and clearly written, effectively communicating complex concepts and findings. The logical flow and organization of ideas contribute to its accessibility and impact.

**Weaknesses:**

The paper's primary limitation lies in the generalizability and effectiveness of its proposed divergence-based attacks. While innovative, several concerns emerge:

1. Limited Applicability:
The prompt-based divergence attack has already been largely addressed by OpenAI and shows limited effectiveness beyond GPT-3.5-Turbo. Similarly, the fine-tuning-based divergence attack demonstrates reduced effectiveness on Llama-2-Chat, suggesting these methods might be model-specific rather than universal.
2. Correlation Concerns:
The relationship between divergence behavior and memorization is not strongly established. The paper would benefit from a deeper analysis of this correlation, as the current results suggest the connection might be specific to OpenAI's training process rather than a general phenomenon across different LLMs.
3. Methodological Limitations:
The heavy reliance on divergence-based attacks as the primary mechanism for revealing memorization might provide an incomplete or potentially misleading picture of the actual memorization behavior.

**Questions:**

- Could the authors elaborate additional analysis in the main text on why the divergence-based attacks show varying effectiveness across different models?

- Have you explored alternative attack methods (beyond divergence-based attacks) that might be more universally effective across different LLMs? I wish to learn the authors' thoughts on this.

- Can the authors provide additional analysis over cases where memorization occurs without divergence?

---

> ### Author Response · Authors · 2024-11-18
>
> Thank you for your review and valuable feedback. We address the specific weakness and questions next:
>
> > The paper's primary limitation lies in the generalizability and effectiveness of its proposed divergence-based attacks.
>
> Our work is the first presenting attacks to extract memorized training data from aligned language models. We introduce a divergence-based attack and a more principled and controllable **fine-tuning attack**. These two are different. The fine-tuning attack reverts the model objective from engaging in conversations back to a pre-training objective of “completing documents”.
>
> As we discuss in the general response, the divergence attack was disclosed to OpenAI and patched before the paper became public. Other model developers have presumably implemented similar mitigations. We do not think this is necessarily bad as the goal of our research is to ultimately protect users.
>
> Our fine-tuning attack is successful in Llama and demonstrates transferability. What is confusing from the results is that overall extractable memorization is not comparable to OpenAI models. If you take a close look at Table 2, the extracted tokens for Llama 2 **before alignment** are below 4% compared to 16% for aligned GPT-3.5 **after attack**. This means that Llama 2 has memorized less text to begin with and we should expect our attack to extract less data overall. Still our attacks are the best known to date and can get over 30% of the memorized text from aligned models.
>
> > Can the authors provide additional analysis over cases where memorization occurs without divergence?
>
> As we report in Figure 2, verbatim regurgitation in aligned models is extremely rare (GPT-3.5-turbo barely outputs such strings). The examples we find for this model are benign sequences like alphabetical lists of countries.
>
> \[1\] Carlini, Nicholas, et al. "Extracting training data from large language models." *30th USENIX Security Symposium (USENIX Security 21\)*. 2021\.

---

> ### Comment · Reviewer_JkUJ · 2024-11-27
> **Thank you for response**
>
> Thank you for your detailed response. I would like to maintain my score of 6.

---

### Author Response · Authors · 2024-11-18
**General comments**

We thank the reviewers for their time and valuable feedback. The reviewers have noted that our paper solves a "critical problem in in LLM development" and that our contributions are "valid and significant." We have addressed the weaknesses pointed out by each reviewer in the comments.

We would like to clarify two central points broadly.

* **Divergence attack does not work on other models because we responsibly disclosed the vulnerability**

We discuss our responsible disclosure in Appendix A.1. We shared our findings on ChatGPT with OpenAI and they requested a 90 days period to fix the issue before we released our work. After this time, the attack was fixed and thus our attack is no longer effective on any OpenAI model and other model providers presumably implemented similar mitigations, as models often refuse to repeat words forever. We do not think this reduces the significance of our research contribution and should not be understood as a limitation of our work.

* **Fine-tuning attack does actually transfer to Llama-2.**

Directly comparing the rates at which we extract memorized tokens from OpenAI and Llama models can be misleading. If we take a closer look at Table 2, we find that the unaligned Llama 2 model only outputs \<4% of memorized tokens. This should be understood as an upperbound for the best attack performance. Our attack is the best to date and can recover over 30% of this memorization from aligned Llama models. This demonstrates that fine-tuning is a generalizable attack across models.

---

### Meta-Review · Area_Chair_pSA3 · 2024-12-11

**Metareview:**

The paper explores the scalable extraction of training data memorization in aligned large language models. With the methods proposed in the paper, the authors successfully extract training samples from different open-sourced and closed-sourced LLMs like ChatGPT and GPT-4. The authors also construct a 10TB text dataset for evaluation, and the results demonstrate the effectiveness of their proposed divergence and fine-tuning attacks. This paper can raise attention on the threats of training data leakage. Therefore, although the proposed methods in this paper are currently slightly "out-of-date" as many new related attacks have been proposed, all the reviewers recommend acceptance of this paper.

Strengths:

1.The problem the paper addresses—training data can be successfully extracted even if the LLMs have been aligned—can raise more attention to the training data leakage problem in both the research and social communities.

2.The paper proposes two effective methods to extract LLM training data from several widely used open-sourced and closed-sourced LLMs. It also proposes metrics and datasets for comprehensive evaluations on data extraction.

Weaknesses:
1.There is a lack of analysis on the reasons for such attacks, which makes the proposed methods seem like occasionally discovered bugs. As a result, the theoretical and technical insights are limited.

2.Many attacks related to or inspired by this paper have already been proposed, which makes this paper slightly "out-of-date."
Main reasons for acceptance: In summary, all the reviewers and I agree that this paper is the first to successfully extract training data from aligned, production LLMs. Its impact and insights to the research and social communities are important. Therefore, all reviewers and I recommend the acceptance of this paper.
However, considering the weaknesses I listed above, I suggest this paper be presented as a poster.

**Additional Comments On Reviewer Discussion:**

In the rebuttal period, the reviewers and authors discussed the work's importance, technical novelty, underlying reasons and etc. All reviewers recommend acceptance of this paper.

---

### Decision · Program_Chairs · 2025-01-22

Accept (Poster)